# Human major infections: Tuberculosis, treponematoses, leprosy—A paleopathological perspective of their evolution

**Maciej Henneberg**[1,2,3]* , **Kara Holloway-Kew**[4] , **Teghan Lucas**[3,5]

**1** Biological and Comparative Anatomy Research Unit, Adelaide Medical School, University of Adelaide, Adelaide, Australia, **2** Institute of Evolutionary Medicine, University of Zurich, Zurich, Switzerland, **3** Department of Archaeology, Flinders University, Adelaide, Australia, **4** Institute for Mental and Physical Health and Clinical Translation, School of Medicine, Deakin University, Geelong, Australia, **5** School of Medical Sciences, Anatomy, University of New South Wales, Sydney, New South Wales, Australia

☯ These authors contributed equally to this work.

* maciej.henneberg@iem.uzh.ch

**Data Availability Statement:** All relevant data are within the manuscript and its Supporting information files.

## Abstract

The key to evolution is reproduction. Pathogens can either kill the human host or can invade the host without causing death, thus ensuring their own survival, reproduction and spread. Tuberculosis, treponematoses and leprosy are widespread chronic infectious diseases whereby the host is not immediately killed. These diseases are examples of the co-evolution of host and pathogen. They can be well studied as the paleopathological record is extensive, spanning over 200 human generations. The paleopathology of each disease has been well documented in the form of published synthetic analyses recording each known case and case frequencies in the samples they were derived from. Here the data from these synthetic analyses were re-analysed to show changes in the prevalence of each disease over time. A total of 69,379 skeletons are included in this study. There was ultimately a decline in the prevalence of each disease over time, this decline was statistically significant (Chi-squared, p<0.001). A trend may start with the increase in the disease's prevalence before the prevalence declines, in tuberculosis the decline is monotonic. Increase in skeletal changes resulting from the respective diseases appears in the initial period of host-disease contact, followed by a decline resulting from co-adaptation that is mutually beneficial for the disease (spread and maintenance of pathogen) and host (less pathological reactions to the infection). Eventually either the host may become immune or tolerant, or the pathogen tends to be commensalic rather than parasitic.

## Introduction

As metazoans, humans host whole ecosystems of microbiota and adapt to them. Many of the small organisms living in the human body, or on its surface, are harmless (commensals) or

**Funding:** The Authors received no specific funding for this work.

**Competing interests:** The Authors have declared that no competing interests exist.

even helpful (symbionts). Some others, however, are harmful (pathogens). These pathogens, like all organisms, are subject to natural selection, and thus aim to maximize their reproductive success. This can be achieved either by shortening their generation time and increasing the number of offspring to the extent that the pathogen will kill the host quickly thus increasing its virulence [1], or by slowing reproduction while reducing virulence or changing its antigenic reactions to ensure its own survival in host tissues, thus, extending its own reproduction time for many generations [2]. In the long run, it is advantageous for the pathogen to not cause the death of its host, thus ensuring the reproductive success of the pathogen [3]. At the same time, survival of the host can be only ensured through reproductive success. Therefore, there are basically only two possible outcomes of an initial interaction of a metazoan organism and an invading pathogen. Either (1) the pathogen is so strongly detrimental to the host's organism that it will kill it quickly, or (2) the host and pathogen become eventually adapted to each other which ends with the pathogen becoming a commensal or even a symbiont [4, 5]. Tuberculosis is an example of this process in progress because only about 5–10% of individuals infected with *Mycobacterium* develop the disease though it has not become a symbiont yet [6].

The first outcome (1) does not support the spread of a pathogen, especially if sick people are isolated from the rest of the community and rules of hygiene are strictly adhered to. The second provides examples of evolutionary change. Since microorganisms have very fast turn-over of generations and microevolution in humans can occur relatively fast, tracing co-evolution of human hosts and pathogens can provide good grounds for the study of evolutionary processes [7–10]. Studying humans, rather than other organisms, as hosts is advantageous because good evidence exists in the form of intentionally buried skeletal remains and, eventually, written records.

Tuberculosis, treponematoses and leprosy are well known infectious diseases in the paleo-pathological record. They have been specifically chosen here to investigate the co-evolution of host and pathogen as they: are widespread in many human populations, have a reasonably well-known pathology (including diagnosing the skeletal signs) and were regularly recorded for at least several centuries. All three are not immediately lethal, and only some of individuals infected show pathological signs in hard tissues (bones and teeth) [11]. This suggests that their pathogens and human hosts have already undergone a part of the co-evolutionary process towards co-adaptation that is trending to commensalism which makes these diseases ideal to investigate coevolution.

Although there were numerous attempts at discerning the origin of these diseases and phylogenetic schemes of pathogens were proposed and discussed [12–18] there is still a need for more work to achieve clarity and consensus. We do not propose here to debate issues of origin and phylogeny, just provide a quantitative study of changes through time in manifestation of the three diseases.

Literary and artistic sources from the past also provide some evidence of the three major infectious diseases depicting changes occurring in soft tissues reaching back as far as Ancient Egypt [19–22]. These, however, are difficult to study quantitatively.

Diseases which manifest on hard tissues can be studied in past populations in a quantitative manner through paleopathology; a relatively new discipline, which has already become well-established providing broad insights into origin of diseases, their etiology and history [23] Its methods have been developed by interdisciplinary collaborations and are now rigorously applied [24]. The manifestation of these diseases is preserved for as long as the skeletal remains persist. Due to the preservation of pathological signs on skeletons, it is possible to trace the process of co-evolution of the three major infectious diseases as far back as specimens have been found.

Though this is a benefit, the drawback is that major systemic infections—tuberculosis, treponemal diseases, leprosy—do not produce bone lesions in all sufferers and that preservation of skeletal materials varies. However, the frequencies of cases of a given disease in skeletal samples can be obtained. These are proxies of the prevalence of a given disease and of its severity, mixed together: the more prevalent the disease was and severe enough to be expressed in pathological changes in hard tissues, the greater the frequency of skeletons bearing its signs among all skeletons in a given sample archaeologically recovered from an old burial site. A previous study [25] has shown a decline in skeletal lesions due to tuberculosis in skeletal remains over a long time period (7250 BCE to 1899 CE) and it would be of interest to determine whether this observation also occurs for other major infections: treponemal diseases and leprosy.

Although the literature does not allow strict epidemiological meta-analyses of palaeopathological cases, there are large studies synthesizing information about manifestations of major chronic infections in skeletal materials that encourage narrative speculative overview of evolutionary changes in their host-pathogen relations. This paper presents such an overview.

## Materials and methods

A large number of paleopathological descriptions of various past skeletal materials have been published. Descriptions of the pathological signs in the published materials were sufficient to make a diagnosis. Various methods were used for diagnosis, these include: ancient DNA, analysis of pathological changes in the skeletal and dental material, microscopic changes in hard tissues. These observations were used in differential diagnoses in peer reviewed papers. In the current paper we compare the three previously published, the fullest to date, synthetic analyses of the three infections in order to find whether frequencies of their manifestations in skeletal samples changed significantly through time on entire continents because such widespread changes can hardly be a result of locally changing environmental conditions. Changes in observed frequencies are a result of natural phenomena and human activities unrelated to purposeful elimination of pathogens from hosts' bodies because until the 20th century there were no effective pharmacological methods to cure these diseases.

### Tuberculosis

Holloway et al. reported all paleopathology cases of tuberculosis, with no limits on geographic location [25]. A total of 531 cases were reported for three different time periods between 7250 BCE to 1899 CE. The three periods as reported by Holloway et al. are pre-urbanised, early urbanised and early modern [25]. The exact dates of these periods vary depending on the geographical location of sites, for a full description of dates see Holloway et al. Among 531 sites reported, there were 91 for which total sample size of the collection of skeletons at a site where they were found was not known [25]. Thus, we could analyse frequencies of 440 cases among the total sample of 43,520 skeletons. Details in the S1 Table and more details in the Appendix A in Holloway et al. [25].

### Treponematoses

Powell and Cook reported all paleopathology cases of treponematoses from America dated from 6000 BCE to the present [26]. The term 'treponematoses' is a broad term used to describe four separate diseases, namely, yaws, pinta, endemic syphilis and venereal syphilis, all forms of treponematoses are considered by Powell and Cook [26]. A total of 775 cases of treponematoses in a total sample of 8717 skeletons were reported for four different time periods. This publication is a contribution of multiple authors with each chapter focussing on all of the reported paleopathology cases of treponematoses in different regions in America. The periods as

defined by Powell and Cook are as follows: 6000 BCE-1000 BCE, 1000 BCE-1000 CE, 1000 CE-1942 CE, 1942 CE-present [26]. All authors which contributed to this publication, have used studies which applied differential diagnoses to identify treponematoses in skeletal samples. Cases from other parts of the world, were subject to debates until very recently and not systematically reported [18]. No specific exclusion criteria were stated for this source other than incomplete data not being reported [26]. The data were obtained from 'demographic profile' summary tables which reported the total number of skeletons analysed at each site and the total number of cases for each site [26]. It is now made clear in paleopathological literature that all diseases included into the term "treponematosis" are caused by the same pathogen (*Treponema pallidum*) despite their phenotypic manifestations that depend on environmental conditions and age at infection [18].

## Leprosy

A study by Schreier reports 1645 paleopathological cases of leprosy among the total of 17,142 skeletons dated between 3125 BCE and 1905 CE as represented by four different periods [27]. The periods are defined as follows: Bronze Age (Pre 600 BCE), Iron Age (500 BCE-1050 CE), Middle Ages (1050 CE-1536 CE) and Early Modern Era (1536 CE-1905 CE). All known cases from the literature were included in the synthetic analysis after removal of duplicate cases and those provided by non peer-reviewed sources [27]. The cases of leprosy were grouped into six geographical regions: Northern Europe, Central and Western Europe, Mediterranean, Asia, Oceania and New World. We, however, run the analysis for the whole world, because continental samples were of insufficient size. In all analyses, the Bronze Age and Iron Age have been combined to increase sample size. Thus, the total number of periods analysed in this study is three. Details of all cases can be found in the S1 Table and in the Appendix A in Schreier [27].

## Statistical analyses

The authors of the synthetic analyses separated their data into specific periods relevant to the history of the disease. In the current paper, midpoint dates of periods were calculated by averaging dates of individual samples included in that period rather than considering simple halfway between "begin" and "end" dates of each period, because distributions of individual samples in some periods were not symmetrical. The dates reported by the synthetic analyses of each of the diseases, were converted to years 'before present' for ease of comparison, in archaeological terms, 'present' is defined as 1950.

LOESS (locally weighted scatterplot *smoothing)* function of SPSS v. 24 was used to fit curves representing changes through time of frequencies of each disease manifestation in skeletal samples. Frequencies in individual samples are subject to large random variations due to various archaeological methods of retrieval of skeletal remains, kinds of burials (eg. primary individual or secondary multiple etc), state of preservation of hard tissues varying depending on taphonomic conditions and human intervention in the past (eg. mummification, placement in stone cysts, sarcophagi etc.) and diagnostic methods applied by researchers who originally described pathologies. Therefore, any detectable directional changes through time, however small, are worth attention and statistical testing. Since no test of the significance of the LOESS curve was available, to provide tests of statistical significance of general direction of trends we have constructed contingency tables. Contingency tables including the number of skeletons showing signs of a disease and the number of skeletons free of pathological changes for each disease in each period were constructed. Significance of observed changes was determined by Chi-squared values (significance level <0.05) and Cramer's V values were used to show the

effect size. Value of Cramer's V, also known as the 'phi' coefficient calculable from contingency tables, is a measure of statistical association similar to correlation coefficients in parametric regressions and thus in the case of this study estimates the portion of variance in disease prevalence explained by the change of time.

## Results

Tuberculosis shows decreasing frequency of skeletal signs over time (Fig 1).

The overall frequency of the skeletal signs of the disease is relatively low, just 1% (Table 1) with a maximum of 27% except for two outliers that have a reported frequency of 100%. However, these are the cases that only have a sample size of two skeletons, likely from sites not fully explored. They are included in the analysis because they meet formal criteria—published reports include sample size. No outliers were removed from the sample arbitrarily. LOESS regression line of frequencies on dates provided by Holloway et al. declines slowly through time with the rate of the decline increasing in the last 2000 years [25]. There is a statistically significant decline of the prevalence of tuberculosis through time, Chi-squared = 37.92, df = 2 (p < 0.0001), Cramer's V = 0.0294 (Table 1).

Treponematoses in America show a slow increase in frequencies towards approximately 3000 years ago, then a period of relative stability followed by a faster decrease in the last 1000 years, approximately (Fig 2).

Again, there are some frequencies of 100%, however, these have no more than 4 skeletons in the sample size, therefore, they are likely outliers, but included here to avoid arbitrary curtailment of data available in the literature. Apart from the outliers, the greatest frequency was reported at 77.2% while the total approaches 9%. The midpoints of the dates for each period, recalculated as years before present, are as follows: 4900 BP, 1742 BP, 612 BP and 349 BP. The total prevalence in each of these periods is 5.75%, 9.06%, 10.41% and 3.91% respectively. There is a statistically significant difference between the prevalence of treponematoses in the four periods: Chi-squared 50.35, df = 3 (p < .0001), Cramer's V = 0.076 (Table 2).

LOESS curve for leprosy over time shows overall decline, with a local peak at the Middle Ages, corresponding to the well-documented period when leprosaria were organised in Europe (Fig 3).

Like in tuberculosis and treponematoses, there are sites with 100% frequency of skeletons showing signs of the disease, however, these are likely outliers. The highest prevalence, disregarding obvious outliers, is 95%. This is very high, especially with a sample size of 635 skeletons. This is most likely the effect of leprosaria having separate cemeteries, thus selectively burying diseased individuals away from the general population. The majority of sites where the presence of leprosy was reported have a prevalence between 10% and 90%. The total frequency approaches 10%. There is a statistically significant difference between the prevalence of leprosy in each of the three defined periods: Chi-squared 865.61 (df = 2, p < .0001). Cramer's V = 0.215 (Table 3).

## Discussion

The data we analysed, although abundant in terms of archaeologically retrieved burials, suffer from likely errors of reporting. The most important among them is the "publication bias" similar to that in the public health reports–if the presence of signs of the disease were not detected in a particular skeletal sample (frequency = 0), this sample is not included in synthetic analyses, simply because no publication regarding this sample mentioned the disease, thus it was not "detectable" in the literature. As examples of this situation may serve large collections of skeletons (N>200), carefully excavated from Medieval burial grounds of Ostrow Lednicki, Poland

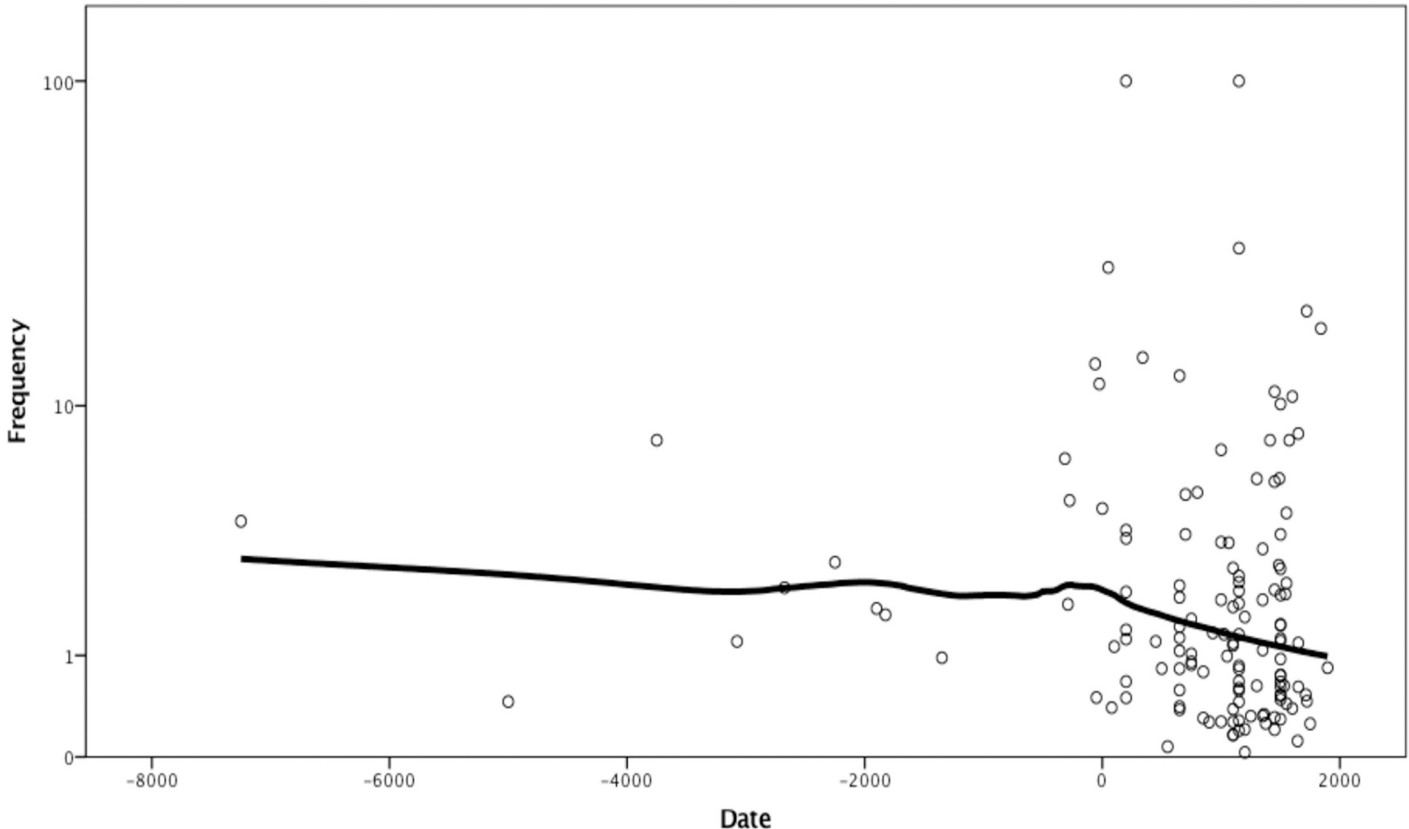

**Fig 1. Logarithmed frequency of skeletal signs of tuberculosis by date.** LOESS curve fitted with 95% of points included and tricube kernel.

[28, 29] or of Classical Antiquity—Metaponto, Italy [30, 31] studied by paleopathologists. In the first one, no signs of leprosy nor tuberculosis were found, in the second, tuberculosis was reported, but leprosy not found. Various burial sites hold skeletons of people who died over longer or shorter periods and from a number of different causes. An extreme example of this is the sample from Pompeii [32] that for paleopathological analyses presents a "survey" of a living population who died within some 16 hours in contrast to samples from burial grounds where practically all individuals died of natural, rather than catastrophic, causes over several centuries. Another source of error is inconsistent application of diagnostic criteria changing through the period in which palaeopathology was practiced and varying among authors. Finally, widespread theoretical "consensus" may have prevented some authors from publishing observations contrary to the prevailing opinion. The recent debate regarding the Columbian origin of

**Table 1. Frequency of bone signs of tuberculosis in consecutive periods of human history.**

| Periods (CE) | Midpoint dates (BP)* | N cases** | N total | Frequency (%) |
|---|---|---|---|---|
| Pre-urbanised | 4700 | 149 | 9593 | 1·55 |
| Early urbanised | 1300 | 161 | 18533 | 0·87 |
| Early modern | 300 | 130 | 15704 | 0·82 |
| TOTAL | | 440 | 43520 | 1·01 |

Chi squared = 37.92 (p < 0.0001), Cramer's V = 0.0294.

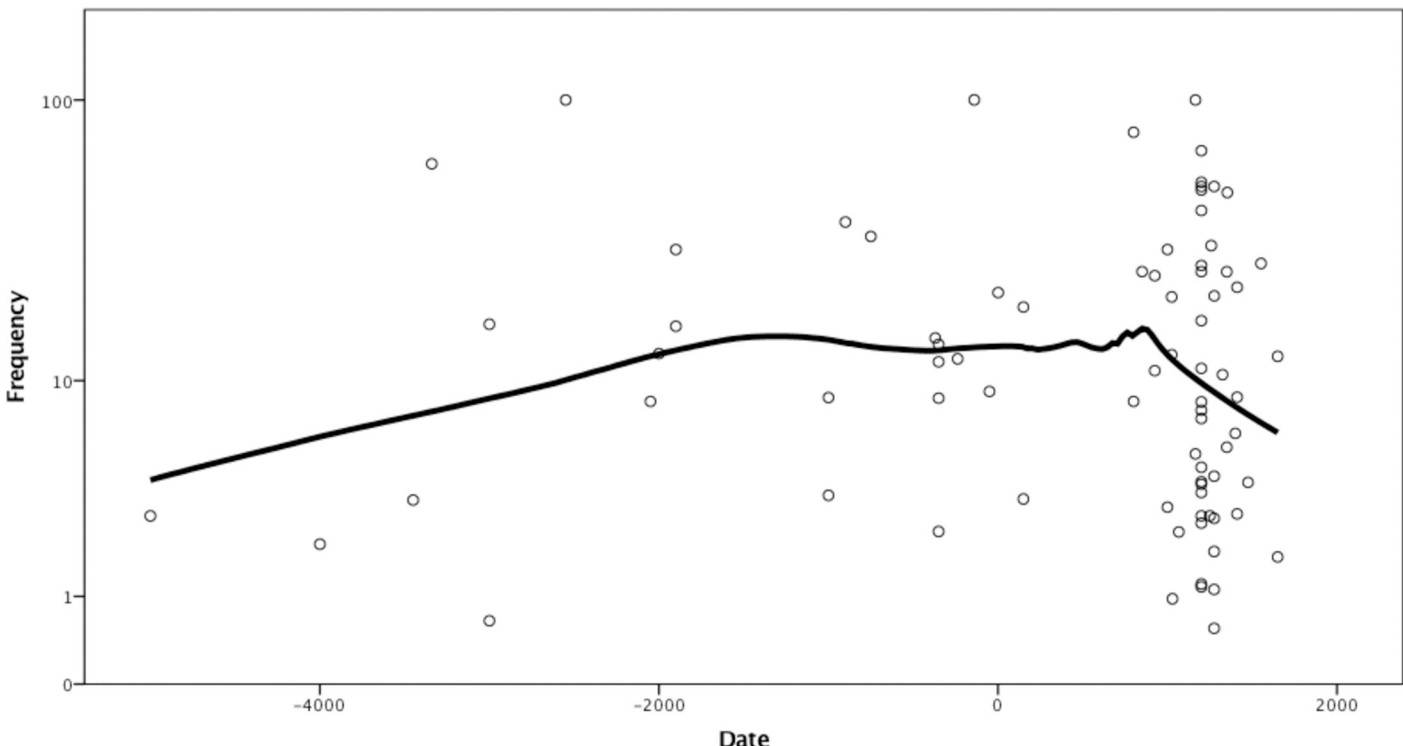

**Fig 2. Logarithmed frequency of skeletal signs of treponematoses by date.** LOESS curve fitted with 70% of points included and tricube kernel. Frequency scale logarithmic.

syphilis in Europe made us study data on the frequency of signs of treponematoses only in America since this continent has no "theoretical bias" which would influence publication of observations [18, 33, 34].

Leprosy has a much higher prevalence in many sites surveyed than do treponematoses and tuberculosis. The reason for this is simple, special establishments—leprosaria existed whereby all deceased affected by the disease would be buried together, away from people who died of other causes. One of the last leprosaria in Europe—Metković. Croatia—was closed as late as 1925 [35].

There are cases where infected persons were buried with non-infected persons, although leprosaria were a common way to prevent the disease from spreading thus concentrating

**Table 2. Frequency of bone signs of treponematosis in consecutive periods of American history.**

| Periods (CE) | Midpoint dates (BP)* | N cases | N total | Frequency (%) |
|---|---|---|---|---|
| -6000 - -1000 | 5500 | 106 | 1844 | 5·75 |
| -1000- +1000 | 1950 | 121 | 1396 | 8·67 |
| 1000–1492 | 700 | 531 | 5102 | 10·41 |
| 1493–1900 | 300 | 17 | 435 | 3·91 |
| TOTAL | | 775 | 8717 | 8·89 |

*rounded to full/mid centuries.

Chi-squared 50.35 (p < 0.0001), Cramer's V = 0.076.

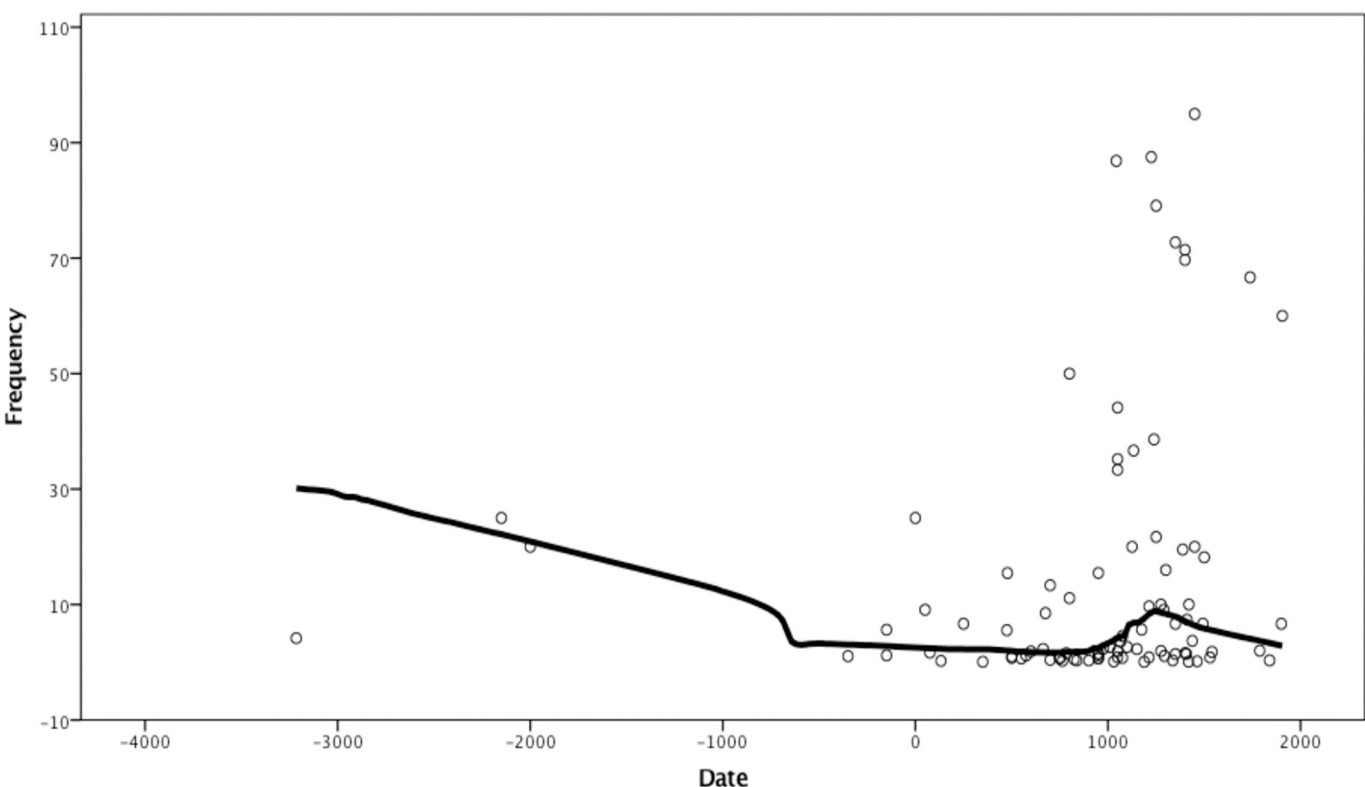

**Fig 3. Frequency of skeletal signs of leprosy by date.** LOESS curve fitted with 60% of points included and tricube kernel.

dying lepers in adjacent cemeteries. Social stigma attached to leprosy contributed to the isolation of patients from the rest of the society and their concentration in leprosaria.

Each of the three diseases shows a decline towards modern times. In treponematosis and in leprosy there is an initial increase in the prevalence of skeletal signs (Tables 2 and 3), while prevalence of skeletal signs of tuberculosis shows a steady decrease (Table 1). It can be postulated that tuberculosis came into contact with humans earlier than the other two diseases and thus experienced a period of increase before the first cases were captured in skeletal samples. Before the advent of agriculture, human groups had low density and burial grounds contained small numbers of individuals, while preservation of older skeletal material is on average less likely than more recent ones and therefore this initial period of tuberculosis increase is not captured by archaeologically recovered skeletal material. Overall frequency of tuberculosis in materials studied is only about 1%, approximately 10 times lower than that of the other two

**Table 3. Frequency of bone signs of leprosy in consecutive periods of human history.**

| Periods (CE) | Midpoint dates (BP)* | N cases | N total | Frequency (%) |
|---|---|---|---|---|
| -6000 – +1050 | 5500 | 158 | 17790 | 0·89 |
| 1050- +1536 | 650 | 1478 | 8931 | 16·55 |
| 1537–1905 | 250 | 9 | 421 | 2·14 |
| TOTAL | | 1645 | 17142 | 9·60 |

*rounded to full/mid centuries.

Chi-squared 865.61 (p < 0.0001) Cramer's V = 0.214.

diseases. This, of course, can be interpreted as a result of different pathophysiology of diseases, but may indicate higher degree of co-adaptation of tuberculosis and humans. Tuberculosis is transmitted by airborne pathogens [36, 37], while leprosy transmission, though possible via airborne droplets, requires close physical contact [38], and treponematoses require body fluid contact to be transmitted thus they could become widespread only when population density increased due, eg. to agriculture. Therefore, only since the advent of agriculture these diseases could spread in human populations evidenced by increase in prevalence and after some time undergo the co-evolution manifesting in decline of their severity.

Frequency of macroscopically observed pathological signs of a given disease in hard tissues is a joint result of the prevalence of the disease and its severity. Both prevalence and severity depend on a number of factors, only one of them being the evolved interaction between host and pathogen. Besides effective therapeutic intervention characteristic for modern medicine, ecological characteristics of past societies such as climate, population density, population distribution and location of population centres, modes of transport, commercial exchanges, nutrition, shelters and clothing, distribution of wealth and, however largely ineffective, early therapeutic practices, certainly had some effects on diseases' prevalence and severity of their signs and symptoms. Influences of all the mentioned factors, and possibly many others, need to be studied in detail. Our broad overview of changes through time, in a large number of skeletal samples representing a variety of ecological, cultural and socio-economic circumstances, and based on frequencies subject to numerous sources of error, and even bias, shows a commonality between manifestations of three different chronic infections having different pathophysiologies and modes of transmission. We suggest that the underlying common factor of these changes is the co-evolution of hosts and pathogens, though it is not possible to indicate which member of the host-pathogen pair underwent greater evolutionary change. What is clear, however, is that adaptation of both has been sufficient to allow them reproductive success. Adaptation can occur only through the process of evolution. It can be achieved by changes in pathogens, in host's immune systems and in host's bodies providing greater tolerance to infections [39].

## Changes in pathogens

Bacteria causing the three studied diseases cannot survive outside of a host's organism, thus they have no environmental reservoirs, which contrasts with many other pathogens. It is important that these bacteria balance both virulence and transmission such that they do not kill the host too quickly to allow enough host's lifetime to facilitate spread [40]. This balance requires ongoing changes in bacterial genetics as a response to host and environmental changes that may affect how well or poorly the bacteria can spread.

Tuberculosis and leprosy have repeating elements in their aDNA, which are used to detect their presence in skeletal samples. For tuberculosis, these elements include IS6110 and IS1081, and for leprosy these are RLEP and REPLEP [41]. There are also genetic variations in these bacteria which allow categorisation into separate groups. For example, "ancient" and "modern" strains of *Mycobacterium tuberculosis* are distinguished by the deletion of a particular section of the genome, known as TbD1; the *M. tuberculosis* specific deletion [42].

Stone and colleagues [43] have reviewed multiple previous studies that have examined the genetic heterogeneity within the group of bacteria that cause tuberculosis. This group is known as the Mycobacterium tuberculosis complex (MTBC) and includes *M. tuberculosis* (humans), *M. africanum* (humans), *M. canetti* (humans), *M. bovis* (cattle), *M. microti* (voles), *M. caprae* (goats) and *M. pinnipedii* (seals and sea lions). Even within these MTBC members, there are multiple sub-groups, with differences in their genomes. Evolutionary theories have

been proposed, stating that the last common ancestor for the MTBC was at least 15,000 years ago, though estimates vary in the literature. It is clear that the MTBC has been present for at least several thousand years, providing a long period for co-evolution to occur.

Unlike M. tuberculosis, leprosy is caused primarily by only one species: Mycobacterium leprae.

It has a smaller genome than in other mycobacteria and a large number of pseudogenes (~27% of genome) and non-coding sequences (~23.5% of genome) [44]. The genetic variation of *M. leprae* has not been as thoroughly studied in the archaeological record as has *M. tuberculosis*. Based on investigation of single nucleotide polymorphisms (SNPs), two evolutionary hypotheses have been proposed; one suggests that the bacteria arose in East Africa, while the other suggests Central Asia.

There is evidence of adaptation to local host and environmental situations for both modern and historical strains of *M. tuberculosis* and *M. leprae*; different strains are found in different geographical areas [41, 45]. Additionally, multiple different strains can occur in the same geographical region at the same time. For example [46] have reported that during the 18[th] and 19[th] centuries in Britain, there were multiple strains of *M. tuberculosis* present and infection with multiple strains at the same time also occurred.

The genome of *M. leprae* has also changed over time, and a high amount of genetic diversity existed across Medieval Europe [47]. Four different strains of leprosy have been discovered during the Early Medieval Period (476–800 CE) and during the High Medieval Period (1000–1250 CE), three different strains have been found at the same archaeological site. One of the strains found in the later period is known to infect red squirrels and it is possible that this was a source of zoonotic infection. It has been reported that the fur of the red squirrel was a trade item during that period and these animals were also sometimes kept as pets.

Ancient DNA extracted from specimens found in different geographical areas and time periods, shows that genetic variations have occurred in the past. One study involved an analysis of samples with lesions typical of tuberculosis from the Abbey of St Mary Graces, London (1350–1538 CE) [48]. The authors examined SNPs in several genes (*rpoB*, *mtp40* and *oxyR*) and performed spacer-oligonucleotide typing (spoligotyping), which is a technique that has been used to differentiate between members and strains of the MTBC. The ancient strains were similar to modern tuberculosis strains, however, it was reported that these ancient strains did not have any mutations conferring antibiotic resistance, as expected. Another study reported analyses of samples from Hungarian mummies located at Vác, showing that lesions typical of tuberculosis, as well as ancient DNA were present in many of the samples [49]. This study examined several other genes including *gyrA* and *katG*, which can be used to differentiate between strains of the MTBC. The results showed that none of the samples were antibiotic resistant, and they belonged to a modern strain of *M. tuberculosis*. Spoligotyping and further genetic analyses were completed for three samples (a mother and two daughters), showing that although the strain was modern, it was not identical with current strains, indicating that genetic changes have occurred since the 18[th] and 19[th] centuries [50]. The study also reported that two different strains were present among these three samples, showing that multiple strains co-existed at the same time. Another study [51] performed genetic analyses for samples derived from Ancient Egypt, which ranged in age from 2050 BCE to 500 BCE. Spoligotyping showed that some of the older samples from 2050–1650 BCE could be *M. africanum* (a human pathogen, not "modern" genetically and present only in Africa today), while samples from later periods were consistent with *M. tuberculosis*. The samples from the later period had the TbD1 deletion, indicating a "modern" strain.

Poor preservation is a limitation of using ancient DNA and observed absence of particular sections of the genome may not be evidence of absence in the original organism. This is

particularly important for tuberculosis because many of the changes in the genome are deletions of specific sections [42]. It is interesting to note that deletions in the tuberculosis bacterial genome have been shown to result in lower virulence and a lower likelihood of pulmonary cavitations [52], but no change in transmission or pathogenicity indices. However, for leprosy, different strains have not been observed to have differences in virulence [47].

No aDNA of the *Treponema pallidum* dating before the Columbus's contact with Americas has been recovered and studied [18].

## Human resistance and tolerance revealed by genetic analyses

Several studies have examined variation in specific immunological genes, showing that some are associated with higher susceptibility or resistance to tuberculosis or leprosy [53–56]. One such example is the SLC11A1 gene (also known as Natural Resistance-Associated Macrophage Protein 1, NRAMP1) which is associated with natural resistance to intracellular pathogens, including tuberculosis and leprosy. Barnes and colleagues [57] have demonstrated a gradient in this allele, where geographical regions that had a longer time since first settlement (range >6500 years ago to modern times) had a higher frequency of the allele compared to those settled more recently. Spigelman and colleagues [15] also describe variation in the SLC11A1 gene, showing that there was diversity in this gene among two different groups: individuals from Hungary (18[th] century) and Sudan (500–1400 CE). The authors reported that some alleles appeared to confer resistance to tuberculosis infection. These studies indicate that there may have been changes in the genetics of the host during early urbanisation. However, SLC11A1 is not the only gene that could have changed over time and these diseases are not the only potential influence for these changes. It is likely that there is a complex response to changing pathogen, host and environmental conditions over time.

## Earlier considerations of co-evolution of the diseases and humans

Some studies have stated that for tuberculosis to exist in its current form, a certain level of population number and density was required [58]. Through time, tuberculosis (and other diseases) gained opportunities for transmission due to an increasing population density and urbanisation. Therefore, it could be expected that through time, tuberculosis would change to become more virulent, allowing a higher rate of transmission in a more densely populated environment. This would then lead to a higher frequency of lesions on skeletal remains. However, this was not observed in this study, indicating some changes in the bacteria and host may have occurred over time, which could include genetic changes [54].

It is accepted that tuberculosis and leprosy have affected humans for several thousand years. Although it is expected that host and pathogen will adapt to each other over time, this is complicated to study and describe [30]. For tuberculosis and leprosy, the manifestation of the disease is dependent upon many factors including the virulence of the pathogen, host immunity, host environment and host tolerance. Host environmental conditions can include nutrition, living conditions, comorbidities and social factors. For example, following industrialisation, a decline in mortality due to tuberculosis was observed during the 19[th] century in many major cities, including London and New York. This decline occurred prior to the introduction of antibiotics and was instead due to a number of changes aimed at controlling and preventing tuberculosis and increasing host's tolerance. These included major sanitation and public health acts, notification of infectious disease requirements, compulsory reporting and development of better diagnostics (e.g. X rays), and improved nutrition [59–63]. For leprosy [64], it has been suggested that it is likely to have appeared during urbanisation, as there is

no skeletal evidence prior to this. In addition, leprosy requires close contact for infection which required a sufficient population density in order to occur effectively.

The co-evolution of tuberculosis and humans over time, as shown in paleopathological and ancient DNA evidence has been discussed [54]. They state that tuberculosis has adapted to states of both high and low transmission. In some cases, the disease is latent and in others it causes pulmonary symptoms which results in rapid transmission in areas with a high population density. Gagneux [40] describes characteristics of "modern" and "ancient" strains of tuberculosis, where more modern strains are more successful in terms of transmission, virulence and have a shorter latency time, which are related to increases in population density over time.

Treponemal diseases are now known to have occurred all over the world for a long time preceding the travel of Columbus to the New World, though their occurrence outside of Americas is not yet synthetically summarised. Their phenotypes differ depending on the environment (eg. bejel in warm dry climates) and the age at primary infection (eg. yaws transmitted in childhood). Studies of ancient DNA and of phylostratigraphy of modern genetic variation of treponemes cannot reach back in time more than a few hundred years, thus no longer-term evolutionary variation of their genomes is known [14, 18].

## Co-infections of tuberculosis, leprosy and treponematosis

There have been examples of co-infection of tuberculosis and leprosy reported in the literature, indicating that ancient populations had to combat the pressures of multiple diseases at once. A clear 19[th] century paleopathological case of co-infection of tuberculosis and treponematosis has been reported, too [65]. In this case, an approximately 10-years old child with signs of congenital syphilis had extensive pathologies of the vertebral column typical for tuberculosis. Weiss and colleagues [66] reported vertebral changes typical of tuberculosis infection in a male skeleton dated to around 1400 CE, buried at the site of a leprosarium. The skeleton also shows skeletal changes indicating leprosy, as well as osteoarthritis and fractures of the clavicle and ribs.

Another study has shown evidence of tuberculosis and leprosy at an Iron Age site in northeastern Thailand, indicating that co-infection was likely, though skeletal signs for both diseases were not present together on the same individuals [67]. Two individuals were described as having skeletal changes consistent with leprosy, while another individual had changes suggesting tuberculosis infection.

Multiple authors have suggested that the historical decline in leprosy was due to an increased mortality due to tuberculosis [41, 59, 68]. This suggestion is supported by the presence of both *M. tuberculosis* and *M. leprae* aDNA in skeletal samples dated to the 13[th] century. Evidence of cross-immunity between the two bacteria exists, however, since many of these skeletons were positive for both tuberculosis and leprosy, this appears unlikely. Additionally, since the effects of leprosy are dependent upon the immune response of the host, it is possible that the lowered immunity of individuals infected with leprosy, accompanied by their poorer living conditions due to stigma, allowed tuberculosis to more easily infect those with leprosy. The increasing mortality in these individuals would have resulted in a decline in the numbers of people with leprosy and ultimately a decline in the disease as a whole.

## Conclusion

This analysis shows that all three chronic and widespread infections, when their skeletal signs are studied in the context of paleopathological records of populations where they occurred, eventually were gradually decreasing their severity. In the last 5000 years skeletal signs of

tuberculosis become less common, skeletal manifestations of leprosy in Europe declined after the end of the Middle Ages, while skeletal signs of treponematoses in North America declined, especially in the last years before the contact with invading Europeans.

## Supporting information

**S1 Table. Data used.** Locations, dating, sample size and numbers of pathological cases for each of the studied diseases.
(XLSX)

## Author Contributions

**Conceptualization:** Maciej Henneberg, Kara Holloway-Kew, Teghan Lucas.

**Formal analysis:** Maciej Henneberg.

**Investigation:** Maciej Henneberg, Kara Holloway-Kew, Teghan Lucas.

**Writing – original draft:** Maciej Henneberg, Kara Holloway-Kew, Teghan Lucas.

**Writing – review & editing:** Maciej Henneberg, Kara Holloway-Kew, Teghan Lucas.

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
