## [Decision Letter · Decision Letter 0]

21 Dec 2020

PONE-D-20-37249

Human major infections: tuberculosis, treponematoses, leprosy – a paleopathological perspective of their evolution

PLOS ONE

Dear Dr. Henneberg,

Thank you for submitting your manuscript to PLOS ONE. After careful consideration, we feel that it has merit but does not fully meet PLOS ONE’s publication criteria as it currently stands. Therefore, we invite you to submit a revised version of the manuscript that addresses the points raised during the review process.

We look forward to receiving your revised manuscript.

Kind regards,

David Caramelli, Ph.D

Academic Editor

PLOS ONE

Journal Requirements:

2.) PLOS requires an ORCID iD for the corresponding author in Editorial Manager on papers submitted after December 6th, 2016. Please ensure that you have an ORCID iD and that it is validated in Editorial Manager. To do this, go to ‘Update my Information’ (in the upper left-hand corner of the main menu), and click on the Fetch/Validate link next to the ORCID field. This will take you to the ORCID site and allow you to create a new iD or authenticate a pre-existing iD in Editorial Manager. Please see the following video for instructions on linking an ORCID iD to your Editorial Manager account: https://www.youtube.com/watch?v=_xcclfuvtxQ

3.) Please include captions for your Supporting Information files at the end of your manuscript, and update any in-text citations to match accordingly. Please see our Supporting Information guidelines for more information: http://journals.plos.org/plosone/s/supporting-information.

4.) Please amend your list of authors on the manuscript to ensure that each author is linked to an affiliation. Authors’ affiliations should reflect the institution where the work was done (if authors moved subsequently, you can also list the new affiliation stating “current affiliation:….” as necessary).

Reviewers' comments:

Reviewer's Responses to Questions

**Comments to the Author**

1. Is the manuscript technically sound, and do the data support the conclusions?

Reviewer #1: Yes

Reviewer #2: Yes

2. Has the statistical analysis been performed appropriately and rigorously? 

Reviewer #1: Yes

Reviewer #2: N/A

3. Have the authors made all data underlying the findings in their manuscript fully available?

Reviewer #1: Yes

Reviewer #2: Yes

4. Is the manuscript presented in an intelligible fashion and written in standard English?

Reviewer #1: Yes

Reviewer #2: Yes

5. Review Comments to the Author

Reviewer #1: Dear Authors,

the manuscript you submitted is well-presented and in my opinion should be accepted. I would, however, recommend some changes in order to improve its overall quality:

- leprosarium/leprosaria is the correct form, not leprosorium/leprosoria. Plus, maybe the authors could mention the social stigma associated with leprosy, evident from the Bible on, since leprosy patients would be expelled from their communities as they were considered cursed by God;

- leprosy, tuberculosis and syphilis can also be investigated in the past by re-assessing artistic sources, which provide insights on their presentations, which is particularly helpful in terms of soft-tissue assessment: some references that could be quoted are:

Varotto E, Ballestriero R. 17th-century sculptural representation of leprosy in Perugia's Cathedral. Infection. 2018 Dec;46(6):893-895. doi: 10.1007/s15010-018-1237-y;

Papa V, Galassi FM, Varotto E. ‘Representation of spinal tuberculosis in a Ptolemaic dwarf statuette’. In: Van Hee R, do Sameiro Barroso M, Galassi FM (eds), special issue of Vesalius, Proceedings of the 46th Congress of the International Society for the History of Medicine ‘Aesculapius in Lisbon’, 3-7 September 2018, 2020, pp. 188-196.

Fornaciari A, Gaeta R, Giuffra V. Leprosy in the Pisan fresco "Triumph of Death" (1336-1341). J Infect. 2018 Jul;77(1):75-81. doi: 10.1016/j.jinf.2018.04.015.

- in the bibliography the authors include (ref. 53) the Buzic and Giuffra paper: I would avoid this reference because the paper is very general and draws on previous papers, not really a good quality one to be included in this venue. I would suggest opting for a more prestigious (genetics-based) paper discussing the evolution of tuberculosis. They could include in their discussion the following paper: https://www.ncbi.nlm.nih.gov/pmc/articles/PMC4550673/

- p. 12 "while leprosy transmission requires close physical contact" ==> this is correct, but airborne transmission via droplets is also described, so I would rephrase the sentence

Reviewer #2: The paper reads well and offers an interesting global analysis of the problems of some of the major infections at the heart of the paleopathological debate.

I think the authors should add more references about the origins of the discussed diseases and recent phylogenetic studies about them. In addition, a couple of general theoretical papers on the paleopathological approaches that could be mentioned in the ref. list are:

Rühli FJ et al. Palaeopathology: Current challenges and medical impact. Clin Anat. 2016 Oct;29(7):816-22. doi: 10.1002/ca.22709.

Buikstra JE, Cook DC, Bolhofner KL. Introduction: Scientific rigor in paleopathology. Int J Paleopathol. 2017 Dec;19:80-87. doi: 10.1016/j.ijpp.2017.08.005.

I would also add some description of the historical role of leprosaria:

Wokaunn M, Jurić I, Vrbica Z. Between stigma and dawn of medicine: the last leprosarium in Croatia. Croat Med J. 2006 Oct;47(5):759-66.

The introduction is correct but a bit long, I think it can be shortened (I would cut 2-3 lines and make it more direct).

I noted some spelling errors here and there, so I would recommend reading the manuscript again carefully.

6. PLOS authors have the option to publish the peer review history of their article (what does this mean?). If published, this will include your full peer review and any attached files.

Reviewer #1: No

Reviewer #2: No

---

## [Author Response · Author response to Decision Letter 0]

26 Jan 2021

THERE WERE NO EDITOR’S COMMENTS. BELOW WE RESPOND TO REVIEWERS’ COMMENTS INSERTING OUR RESPONSES IN CAPITALS (FOR EASY DISTINCTION) AFTER EACH COMMENT. WE ARE GRATEFUL TO REVIEWERS FOR THEIR HELPFUL COMMENTS.

Reviewer #1: Dear Authors,

the manuscript you submitted is well-presented and in my opinion should be accepted. I would, however, recommend some changes in order to improve its overall quality:

- leprosarium/leprosaria is the correct form, not leprosorium/leprosoria. 

WE HAVE CORRECTED SPELLING OF LEPROSARIA THROUGHOUT

Plus, maybe the authors could mention the social stigma associated with leprosy, evident from the Bible on, since leprosy patients would be expelled from their communities as they were considered cursed by God;

WE HAVE ADDED A SENTENCE ABOUT THE SOCIAL STIGMA IN THE DICUSSION (LINE 491-2)

- leprosy, tuberculosis and syphilis can also be investigated in the past by re-assessing artistic sources, which provide insights on their presentations, which is particularly helpful in terms of soft-tissue assessment: some references that could be quoted are:

Varotto E, Ballestriero R. 17th-century sculptural representation of leprosy in Perugia's Cathedral. Infection. 2018 Dec;46(6):893-895. doi: 10.1007/s15010-018-1237-y;

Papa V, Galassi FM, Varotto E. ‘Representation of spinal tuberculosis in a Ptolemaic dwarf statuette’. In: Van Hee R, do Sameiro Barroso M, Galassi FM (eds), special issue of Vesalius, Proceedings of the 46th Congress of the International Society for the History of Medicine ‘Aesculapius in Lisbon’, 3-7 September 2018, 2020, pp. 188-196.

Fornaciari A, Gaeta R, Giuffra V. Leprosy in the Pisan fresco "Triumph of Death" (1336-1341). J Infect. 2018 Jul;77(1):75-81. doi: 10.1016/j.jinf.2018.04.015.

WE HAVE ADDED TEXT ABOUT ARTISTIC SOURCES , CITED REFERENCES SUGGESTED, AND ONE MORE REFERENCE. LINES 160-163.

- in the bibliography the authors include (ref. 53) the Buzic and Giuffra paper:

WE HAVE REMOVED THE BUZIC AND GIUFFRA REFERENCE

 I would avoid this reference because the paper is very general and draws on previous papers, not really a good quality one to be included in this venue. I would suggest opting for a more prestigious (genetics-based) paper discussing the evolution of tuberculosis. They could include in their discussion the following paper: https://www.ncbi.nlm.nih.gov/pmc/articles/PMC4550673/

WE INCLUDED MORE GENERAL, THAN THE ONE SUGGESTED, REFERENCES ON THE GENETICS OF TUBERCULOSIS (NUMBERS IN OUR LIST OF REFERENCES 15 (SPIGELMANN ET AL.), 16 (GAGNEUX ) 

- p. 12 "while leprosy transmission requires close physical contact" ==> this is correct, but airborne transmission via droplets is also described, so I would rephrase the sentence

REPHRASED, LINES 520-1 

Reviewer #2: The paper reads well and offers an interesting global analysis of the problems of some of the major infections at the heart of the paleopathological debate.

I think the authors should add more references about the origins of the discussed diseases and recent phylogenetic studies about them.

RFERENCES ON THE ORIGIN AND PHYLOGENIES WERE ADDED (ITEMS 12-18) IN A TEXT IN LINES 160-163

 In addition, a couple of general theoretical papers on the paleopathological approaches that could be mentioned in the ref. list are:

Rühli FJ et al. Palaeopathology: Current challenges and medical impact. Clin Anat. 2016 Oct;29(7):816-22. doi: 10.1002/ca.22709.

Buikstra JE, Cook DC, Bolhofner KL. Introduction: Scientific rigor in paleopathology. Int J Paleopathol. 2017 Dec;19:80-87. doi: 10.1016/j.ijpp.2017.08.005.

I would also add some description of the historical role of leprosaria:

Wokaunn M, Jurić I, Vrbica Z. Between stigma and dawn of medicine: the last leprosarium in Croatia. Croat Med J. 2006 Oct;47(5):759-66.

ALL THE ABOVE REFERENCES WERE ADDED: numbers 23, 24 and 35 IN THE LIST OF REFERENCES. AND MENTIONED IN TEXT IN PARAGRAPHS RELEVANT TO THEIR CONTENTS

The introduction is correct but a bit long, I think it can be shortened (I would cut 2-3 lines and make it more direct).

WE HAVE CUT THE FIRST LINE OF THE INTRODUCTION, BUT THEN, SINCE REVIEWERS SUGGESTED VARIOUS ADDITIONS, WE COULD NOT SHORTEN THE INTRODUCTION MORE

I noted some spelling errors here and there, so I would recommend reading the manuscript again carefully.

ALL THREE AUTHORS READ THE MANUSCRIPT AGAIN AND CORRECTED SPELLING ERRORS. TWO OF THE AUTHORS ARE NATIVE ENGLISH SPEAKERS.

---

## [Editor Report · Decision Letter 1]

1 Feb 2021

Human major infections: tuberculosis, treponematoses, leprosy – a paleopathological perspective of their evolution

PONE-D-20-37249R1

Dear Dr. Henneberg,

We’re pleased to inform you that your manuscript has been judged scientifically suitable for publication and will be formally accepted for publication once it meets all outstanding technical requirements.

Kind regards,

David Caramelli, Ph.D

Academic Editor

PLOS ONE
---

## [Editor Report · Acceptance letter]

3 Feb 2021

PONE-D-20-37249R1 

Human major infections: tuberculosis, treponematoses, leprosy – a paleopathological perspective of their evolution 

Dear Dr. Henneberg:

I'm pleased to inform you that your manuscript has been deemed suitable for publication in PLOS ONE. Congratulations! Your manuscript is now with our production department. 

Kind regards, 

on behalf of

Professor David Caramelli 

Academic Editor

PLOS ONE